# Effectiveness of complex behaviour change interventions tested in randomised controlled trials for people with multiple long-term conditions (M-LTCs): systematic review with meta-analysis

Tasmin Alanna Rookes [1] , Danielle Nimmons,[1] Rachael Frost [1] , Megan Armstrong,[2] Laura Davies,[1] Jamie Ross,[2] Jane Hopkins,[3] Manoj Mistry,[4] Stephanie Taylor,[2] Kate Walters [1]

¹Research Department of Primary Care and Population Health, University College London, London, UK
²Wolfson Institute of Population Health, Queen Mary University of London, London, UK
³Public Contributor, London, UK
⁴Public Contributor, Manchester, UK

**Correspondence to**
Ms Tasmin Alanna Rookes;
t.rookes@ucl.ac.uk

## ABSTRACT

**Introduction** The prevalence of multiple long-term conditions (M-LTCs) increases as adults age and impacts quality of life and health outcomes. To help people manage these conditions, complex behaviour change interventions are used, often based on research conducted in those with single LTCs. However, the needs of those with M-LTCs can differ due to complex health decision-making and engagement with multiple health and care teams.

**Objectives** The aim of this review is to identify whether current interventions are effective for people living with M-LTCs, and which outcomes are most appropriate to detect this change.

**Methods** Five databases (MEDLINE, Embase, PsycINFO, CINAHL and Web of Science) were systematically searched, between January 1999 and January 2022, to identify randomised controlled trials evaluating effectiveness of behaviour change interventions in people with M-LTCs. Intervention characteristics, intervention effectiveness and outcome measures were meta-analysed and narratively synthesised.

**Results** 53 eligible articles were included. Emotional well-being and psychological distress (depression and anxiety) outcomes were most amenable to change (emotional well-being: standardised mean difference (SMD) 0.31 (95% CI 0.04 to 0.58); depression psychological distress: SMD −0.45 (95% CI −0.73 to −0.16); anxiety psychological distress: SMD −0.14 (95% CI −0.28 to 0.00)), particularly for interventions with a collaborative care approach. Interventions targeting those with a physical and mental health condition and those with cognitive and/or behavioural activation approach saw larger reductions in psychological distress outcomes. Interventions that lasted for longer than 6 months significantly improved the widest variety of outcomes.

**Conclusion** Complex interventions can be successfully delivered to those with M-LTCs. These are most effective at reducing psychological distress in those with physical and mental LTCs. Further research is needed to identify the effective components of interventions for people with two or more physical LTCs and which outcome is most appropriate for detecting this change.

## STRENGTHS AND LIMITATIONS OF THIS STUDY

⇒ Broad search terms across many key databases have enabled a summary of a large amount of evidence exploring the effectiveness of complex behaviour change interventions in people living with multiple long-term conditions (M-LTCs) in a systematic review and meta-analysis.

⇒ These findings impact the development and implementation of such interventions in people with M-LTCs, which is a growing area of research and clinical practice.

⇒ Double screening of eligible papers was only completed for a percentage of the identified studies, meaning some eligible papers could have been missed.

⇒ Complex interventions are difficult to define and determining eligibility can be a challenge so a conservative approach to exclusion was used.

⇒ Some meta-analyses results had high heterogeneity, suggesting that they may not have been comparable.

## INTRODUCTION

Living with multiple long-term conditions (M-LTCs) is defined as an individual having two or more health conditions at the same time. Globally, the presence of M-LTCs is reported to be 27.2% in all adults, rising to 67.0% in those aged 74 and over.[1] With an ageing population, this is predicted to rise in those over 65 in the UK, from 54% in 2015 to 68% in 2035.[2] In comparison with the general population, those with M-LTCs are found to have lower quality of life and higher mortality risk,[3 4] as well as greater healthcare costs and utilisation.[5] As prevalence increases, nationally and globally, the pressure on healthcare services, people living with the conditions, and their carers also increases, making the

challenge of managing M-LTCs a priority for current research agendas and healthcare systems, such as the NHS and the James Lind Alliance.[1 6 7]

Despite this increased focus on managing M-LTCs, many of the beneficial complex interventions designed to help people with M-LTCs are based on findings from research and interventions aimed at people with single health conditions and applied more widely to those with M-LTCs. Complex interventions contain multiple interacting components and often depend on the behaviours of those delivering and receiving the intervention and need to be adapted to the specific participant group and setting.[8] The chronic disease self-management programme has found mixed effectiveness in people with M-LTCs, depending on the combination and type of health conditions participants experience, being more effective for people with a physical health condition and probable depression.[9 10] In a recent systematic review and meta-analysis of 14 studies testing behavioural interventions in people with M-LTCs, there was little to no effect on physical activity and weight loss.[11] Managing M-LTCs presents a different challenge to those with single LTCs due to the often-fragmented care across multiple primary and secondary care teams and the varied and competing needs of the individual.[12 13] These needs include complex priority setting and decision-making to adhere to multiple healthcare regimens and weighing up conflicting advice from different specialties.[14] This is reflected in poorer outcomes when receiving interventions to help with the management of their M-LTCs, suggesting that the specific needs of these people are not being addressed by current interventions.[15]

Systematic reviews highlight that the effectiveness of complex interventions varies depending on the combination of LTCs, the type of intervention being used and the outcome measure being used to assess effectiveness.[9 15] Determining the most appropriate outcome measure is particularly difficult in this population, given the wide variety of outcomes that may be relevant depending on the conditions and challenges faced by the population, for example, clinical outcomes, well-being, quality of life, and distress. Therefore, it is essential we first identify whether complex interventions currently being tested in people with M-LTCs are effective for this population and then explore which outcome measure is most sensitive to detecting a significant positive change in outcomes.

## Aims
1. To evaluate the evidence around the effectiveness of behaviour change interventions in people with M-LTCs.
2. To identify the outcome measures most sensitive to change, to identify the effectiveness of complex behaviour change interventions in people with M-LTCs.

## METHODS
This review is reported in line with the Preferred Reporting Items for Systematic Reviews and Meta-Analyses guidelines[16] (online supplemental table 1) and registered on PROSPERO (ID: CRD42021287847).

## Search
The databases MEDLINE, Embase, PsycINFO, CINAHL and Web of Science were systematically searched from January 1999 to January 2022, and an example search term list can be seen in online supplemental figure 1. The cut-off date of 1999 was chosen, to align with the oldest dated paper identified in a previous Cochrane systematic review for complex interventions in people with M-LTCs[13] and to ensure that the findings were applicable to current healthcare settings. The identified articles were exported into EndNote and duplicates removed. 10% of identified titles and abstracts were independently screened by two reviewers (TAR and DN) against the inclusion criteria, using Rayyan.[17] As there was less than 5% conflict (n=41/938; 4.37%), single screening was conducted by TAR for the remaining titles and abstracts. For full-text screening, 20% were independently reviewed by TAR and DN, and again, there was only a 5% discrepancy (n=2/40; 5%), and so the remainder were independently screened by TAR for eligibility.

## Inclusion criteria
Participants: People with two or more physical and/or mental health conditions.

Intervention: A complex behaviour change intervention to help with the management of health conditions aiming to improve individual health outcomes, which could be targeted at the individual or organisational level.

Comparator: Treatment as usual.

Outcomes: Any individual health outcomes, such as quality of life, emotional well-being, clinical outcomes, psychological distress, behaviour change and pain.

Setting: Interventions delivered in primary and community care settings.

Time frame: Published between 1 January 1999 and 17 January 2022.

Design: Randomised controlled trials.

## Data extraction and outcome measures
Study characteristics were extracted by LD and checked by TAR for completeness. Outcome measures were extracted by TAR and checked by DN for completeness. For all studies, we extracted trial authors, year of publication, country, intervention description and length, study follow-up length, M-LTCs targeted, sample size at each time point, means and SD for relevant outcome measures. Outcome measures were grouped into six categories: (1) quality of life, (2) clinical endpoints, (3) behaviour change, (4) pain, (5) well-being and (6) psychological distress. Where data were not reported in the paper, we contacted the authors for further details (11 approached, 1 responded by providing further data). For the 10 who did not respond, and for outcomes which could not be synthesised using meta-analysis, the available data were summarised using narrative synthesis.

## Study quality assessment

Trial result quality was assessed using the Risk of Bias 2 tool.[18] Initial assessment for 20% of eligible studies (n=12) was completed independently by two reviewers (TAR and DN), with an agreement for 8/12 trials (75%). Therefore, a further 10% (n=6) were independently reviewed, with 100% agreement between the two reviewers. The remaining 38 trial findings were assessed for risk of bias by TAR. Risk of bias was used to inform narrative synthesis but not to exclude studies.

## Patient and public involvement

Two public coapplicants living with M-LTCs were involved throughout the whole project. This included the review process, refining the aims and inclusion criteria, selecting the most important outcomes to focus on, analysing and providing feedback of findings, and in academic and lay dissemination of findings. They have been included as coauthors. A public engagement workshop, involving the two public coapplicants and an additional six people living with M-LTCs, was held to discuss relevant outcomes to focus on, effective ways to disseminate research findings with the public and to develop future research questions.

## Data analysis

Where there were three or more studies reporting the same outcome, we conducted a meta-analysis. RevMan V.5.4[19] was used to calculate standardised mean differences (SMD) and 95% CIs. SMDs were used due to varying outcome measures being used to measure the same outcomes. To decide which outcome measures to pool together under outcome categories, three reviewers (TAR, DN and RF) met and discussed the appropriateness of each measure in the analyses. If there was more than one measures for the same outcome our decision on which to include was based on if one was the primary outcome, the measure most closely related to the outcome and the extent of missing data. Psychological distress was separated into anxiety and depression, as many studies reported both outcomes and we felt the two constructs differed.

Inverse-variance random effects meta-analysis models were chosen, a priori, as the types of interventions and populations studied were expected to be somewhat heterogeneous due to the diverse populations living with M-LTCs and the different approaches and settings of interventions. To explore this expected variation, preplanned subgroup analyses were conducted, including types of interventions, length of intervention. Post hoc analysis of the combination of LTCs (physical-physical, physical-mental and mental-mental) were explored following discussions with public contributors. Interventions were grouped into three intervention types through discussions between three reviewers (TAR, DN and RF), which are well established in the literature, self-management, collaborative care, and cognitive and/or behavioural activation, based on their predominant features (table 1). Exploratory supplementary subgroup analyses were carried out to explore (1) intervention type: self-management, collaborative care, and cognitive and/or behavioural activation, (2) length of intervention: <3 months, 3–6 months and 6+ months, for outcomes immediately postintervention delivery, (3) maintenance of effect: time since the intervention was complete to final follow-up, grouped as <3 months, 3+ months postintervention and (4) type of M-LTCs combinations: physical-physical, mental-physical and mental-mental.

Heterogeneity was assessed across studies using $I^2$ statistics, with proportions greater than 25%, 50% and 75% considered to have low, moderate and high heterogeneity, respectively.[20]

## RESULTS

After searching and deduplication, 9354 titles and abstracts were screened. 200 full-text articles were screened and 53 were included in the synthesis (see figure 1 and online supplemental figure 2).

## Study characteristics

Across 53 randomised controlled trials, there were a total of 14 740 participants with study sizes ranging from 25 to 3324. The studies were published between 2001 and 2021, with a steady increase in publications from 2009 to the present day, with 58% published since 2017. Studies

**Table 1** Example interventions and their characteristics for allocation into the three types of intervention for supplementary subgroup analysis

| Intervention type | Example of intervention and characteristics |
|---|---|
| Self-management | Psychoeducational interventions, with goal setting and problem solving, to help people manage the symptoms and improve their health outcomes related to their health conditions. Also included telemonitoring approaches, where participants received information and had their symptoms monitored over time. |
| Collaborative care | Interventions with behaviour change components which target the participants, the clinical team and the system they are embedded within. Also included stepped care, where participants started with one treatment pathway and depending on progress, could be stepped up to more intensive treatment. |
| Cognitive and/ or behavioural activation | Often use motivational interviewing, cognitive–behavioural therapy, behavioural activation and/or problem solving to empower people to help people understand the association between health conditions and cognitive processes and create healthier behaviours for future problems. |

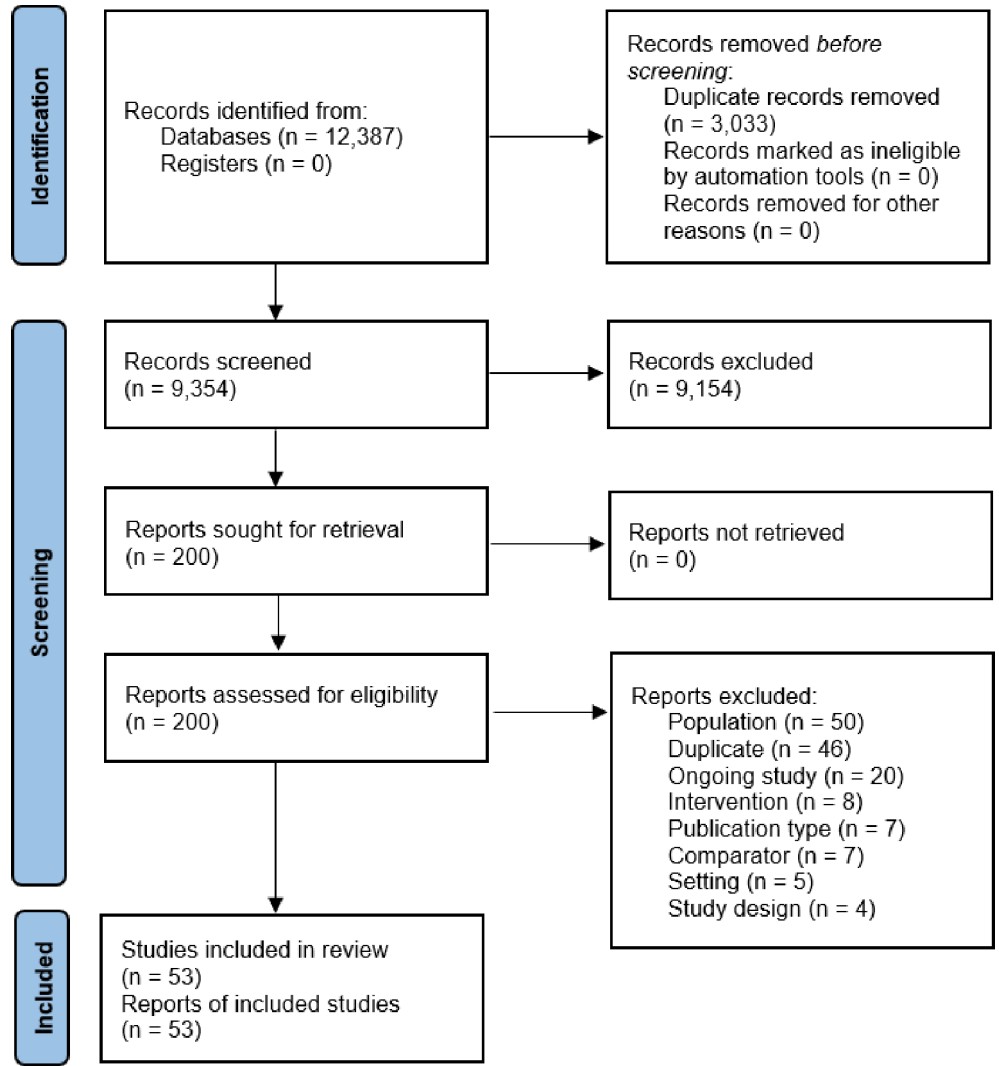

**Figure 1** Prisma flow chart.[16] PRISMA, Preferred Reporting Items for Systematic Reviews and Meta-Analyses.

were conducted in the USA (n=22), Canada (n=7), Australia (n=5), the UK (n=5), Netherlands (n=3), Spain (n=2), Ireland (n=2) and the remaining eight in Singapore, China, Nigeria, Germany, Indonesia, South Korea, Croatia and India (one study conducted across two countries). Interventions were aimed at people with a combination of physical and mental health conditions (n=27), two or more physical health conditions (n=25) and one study for people with a combination of two mental health conditions. Health conditions included diabetes, depression, schizophrenia, coronary heart disease, chronic pain, HIV and hypertension, as well as interventions which were open to people with multiple different health conditions.

Most interventions were self-management interventions (n=28), with 15 collaborative care interventions and 10 cognitive and/or behavioural activation interventions. Interventions were heterogeneous in terms of the person delivering them; by individuals or combinations of nurses, doctors, psychologists, other healthcare professionals and trained peer support leaders. The mode of delivery varied from face to face, in clinics or homes, telephone and through digital platforms. The length of these interventions ranged from 3 weeks to 2 years, with most lasting between 3 and 12 months. Postintervention, maintenance follow-up lengths varied from 6 weeks to 12 months. A summary of the studies and the interventions can be found in online supplemental table 2.

### Risk of bias

Judgement of the overall risk of bias of the trial findings was variable, 29 results were rated as low risk, 20 as having some concerns and four as having a high risk of bias. Of all the studies measuring quality of life and pain, half of the studies measuring clinical endpoints, 80% measuring functioning and 40% of those measuring psychological distress had at least some concerns with risk of bias (see online supplemental table 2).

### Outcome characteristics

For the primary outcome, 10 measured a clinical endpoint relevant to the included LTCs (eg, HbA1c, blood pressure, cholesterol), 10 studies used a psychological distress measure, 5 used a behaviour change measure (eg, alcohol use), 5 a functioning measure (eg, physical

**Table 2** Summary of pooled effects of outcome measures, for each outcome, assessed immediately postintervention and at a follow-up/maintenance time point

| Outcome | Overall effect | | | | | |
| | Postintervention | | | Maintenance | | |
| | n | SMD (95% CI) | I² | n | SMD (95% CI) | I² |
| --- | --- | --- | --- | --- | --- | --- |
| Quality of life | 8 | 0.13 (-0.04 to 0.30) | 59% | 3 | 0.61 (−1.54 to 2.75) | 99% |
| Clinical endpoints | 11 | −0.22 (−0.47 to 0.03) | 88% | 5 | −0.16 (−0.36 to 0.04) | 53% |
| Behaviour change | 6 | 0.06 (−0.18 to 0.29) | 78% | 3 | 0.02 (−0.10 to 0.15) | 23% |
| Pain | 4 | −0.32 (−0.70 to 0.07) | 77% | 0 | N/A | N/A |
| Functioning | 14 | 0.05 (−0.06 to 0.15) | 42% | 8 | 0.05 (−0.05 to 0.16) | 16% |
| Psychological distress (depression) | 21 | **−0.45 (−0.73 to -0.16)** | 95% | 8 | 0.02 (−0.33 to 0.36) | 94% |
| Psychological distress (anxiety) | 9 | **−0.14 (−0.28 to 0.00)** | 40% | 4 | −0.08 (−0.18 to 0.02) | 0% |
| Emotional well-being | 7 | **0.31 (0.04 to 0.58)** | 85% | 7 | **0.28 (0.07 to 0.49)** | 72% |

n is number of studies measuring the outcome. Positive results for psychological distress, indicated by a decrease in scores and for emotional well-being by an increase in scores.
N/A, not available; SMD, standardised mean difference.

functioning), 4 studies measured quality of life, 1 study measured pain, no studies used emotional well-being as the primary outcome and 6 studies used a composite of these measures (often psychological distress combined with a clinical endpoint) and 12 studies used an outcome measure that did not fit into these categories, for example, acceptability of intervention.

For the secondary outcomes included in the meta-analyses, 11 studies measured functioning and psychological distress as an outcome, 9 measured emotional well-being, 7 measured quality of life, 6 behaviour change, and 4 clinical endpoint outcomes.

### Main analysis: meta-analysis of individual outcomes

Outcome data (means and SD for each group) assessed immediately postintervention was available from 31/53 studies for meta-analysis across the seven outcome types. The effectiveness of interventions was detected when emotional well-being (SMD 0.31 (95% CI 0.04 to 0.58)) and psychological distress, both depression (SMD −0.45 (95% CI −0.73 to −0.16)) and anxiety (SMD −0.14 (95% CI −0.28 to 0.00)), were measured. The effect sizes for psychological distress (depression) and emotional well-being (both postintervention and maintenance) were moderate, but heterogeneity was high. For psychological distress (anxiety), the effect size was small and heterogeneity moderate. No other outcomes detected a significant effect from the interventions (see table 2 and online supplemental figure 3 for forest plots).

Outcome data assessed at follow-up, to measure the effect of maintenance, was available for meta-analysis from 15 studies in total across outcomes. The effectiveness of interventions was only detected when emotional well-being was measured (SMD 0.28 (95% CI 0.07 to 0.49)), with moderate heterogeneity. All other outcomes did not detect a significant effect (see table 2 and online supplemental figure 3 for forest plots).

### Narrative synthesis

For the 22 studies, where meta-analysis was not possible due to data not being available in the required format, 15 reported at least 1 of the 7 outcomes and found results mostly consistent with the meta-analysis. Seven studies found positive intervention effects, which measured psychological distress (n=3), clinical endpoints (n=2), behaviour change (n=1) and functioning (n=1) (see online supplemental table 3). Studies measuring quality of life (n=2) found no effects.

### Subgroup analyses

We conducted three supplementary subgroup analyses to explore the heterogeneity within the results by intervention type, duration of intervention and MLTC combination type:

### Intervention type

For interventions with a self-management approach, none of the outcomes detected a significant benefit immediately postintervention and only emotional well-being at follow-up significantly improved (SMD 0.34 (95% CI 0.03 to 0.65)), but high heterogeneity remained for most outcomes (see table 3 and online supplemental figure 3 for forest plots).

For interventions with a collaborative care approach, significant differences postintervention were detected for quality of life (SMD 0.15 (95% CI 0.02 to 0.27)), with low heterogeneity and psychological distress (depression; SMD −0.82 (95% CI −1.40 to −0.24)), with high heterogeneity (table 3). This high heterogeneity may be due to the different outcome measures used to measure depression across the studies, whereas quality of life measures were similar. When measuring maintenance effects, only one outcome was measured by enough studies to conduct meta-analyses and no effects were found.

**Table 3** Pooled effects of outcomes from meta-analysis, stratified by the three groups of complex behaviour change intervention types described in table 1

| Postintervention | Self-management | | | Collaborative care | | | Cognitive and/or behavioural activation | | |
|---|---|---|---|---|---|---|---|---|---|
| | n | SMD (95% CI) | I² | n | SMD (95% CI) | I² | n | SMD (95% CI) | I² |
| Quality of life | 2 | N/A | N/A | 4 | **0.15 (0.02 to 0.27)** | 31% | 2 | N/A | N/A |
| Clinical endpoints | 3 | −0.29 (−0.61 to 0.02) | 61% | 4 | −0.45 (−1.00 to 0.11) | 95% | 4 | 0.06 (−0.10 to 0.23) | 0% |
| Behaviour change | 3 | 0.01 (−0.60 to 0.61) | 91% | 1 | N/A | N/A | 2 | N/A | N/A |
| Functioning | 9 | 0.05 (−0.06 to 0.16) | 26% | 3 | 0.08 (−0.20 to 0.35) | 74% | 2 | N/A | N/A |
| Psychological distress (depression) | 8 | −0.30 (−0.91 to 0.31) | 96% | 6 | **−0.82 (−1.40 to −0.24)** | 97% | 7 | **−0.24 (−0.38 to −0.09)** | 0% |
| Psychological distress (anxiety) | 5 | −0.05 (−0.23 to 0.14) | 0% | 2 | N/A | N/A | 2 | N/A | N/A |
| Emotional well-being | 5 | 0.33 (−0.10 to 0.76) | 90% | 1 | N/A | N/A | 1 | N/A | N/A |
| Maintenance | | | | | | | | | |
| Clinical endpoints | 1 | N/A | N/A | 3 | −0.18 (−0.48 to 0.12) | 74% | 1 | N/A | N/A |
| Functioning | 5 | 0.10 (−0.02 to 0.22) | 0% | 2 | N/A | N/A | 1 | N/A | N/A |
| Psychological distress (depression) | 4 | 0.28 (−0.48 to 1.04) | 96% | 2 | N/A | N/A | 2 | N/A | N/A |
| Emotional well-being | 5 | **0.34 (0.03 to 0.65)** | 76% | 2 | N/A | N/A | 0 | N/A | N/A |

n is number of studies measuring the outcome. P value is considered significant if ≤0.05.
Positive results for psychological distress, indicated by a decrease in scores and for emotional well-being and quality of life by an increase in scores.
N/A, not available; SMD, standardised mean difference.

For interventions with a cognitive and/or behavioural activation approach, postintervention effects were seen for psychological distress (depression) only (SMD −0.24 (95% CI −0.38 to −0.09)), with no heterogeneity.

In the 22 studies synthesised narratively, 2 of the 9 self-management interventions (22%), and 4 of the 6 collaborative care interventions (67%) found significant effects on outcomes, 3 of which were psychological distress (depression) measures. These findings generally support the meta-analysis conclusions (see online supplemental table 3).

### Duration of intervention

For interventions lasting for less than 3 months, positive effects postintervention were only detected with psychological distress (depression) measures (SMD −0.36 (95% CI −0.57 to −0.14)), with low heterogeneity. Maintenance effects could not be explored, as not enough studies measured any of the outcomes.

Significant postintervention effects for interventions lasting between 3 and 6 months were not found for any of the outcomes. However, significant differences at maintenance were detected when emotional well-being was measured (SMD 0.28 (95% CI 0.02 to 0.53)), but heterogeneity was high.

For interventions lasting for longer than 6 months, effects were seen postintervention for clinical endpoints (SMD −0.15 (95% CI −0.28 to −0.02)), with low heterogeneity and psychological distress (depression; SMD 0.50 (95% CI −0.69 to −0.30)) measures, with moderate heterogeneity (see table 4). Maintenance effects in interventions lasting longer than 6 months were only identified with psychological distress (depression) measures (SMD −0.28 (95% CI −0.46 to −0.10)), but heterogeneity was high (table 4 and online supplemental figure 3 for forest plots).

Exploring the 22 studies without data for meta-analysis, none of the 3 studies lasting less than 3 months, 2 of the 5 studies lasting between 3 and 6 months (40%), and 4 out of the 7 studies lasting longer than 6 months (57%) had significant primary outcome effects (see online supplemental table 3).

### M-LTCs combination type

Analysis around the effectiveness of interventions depending on the combination of M-LTCs types, grouped as physical-physical, physical-mental and mental-mental was explored post hoc. The only significant effect found was postintervention for psychological distress (depression) measures in people with combined physical and mental health conditions (SMD −0.58 (95% CI −0.94 to −0.21)), but heterogeneity was high (see online supplemental table 4 and online supplemental figure 3 for forest plots).

In the narrative synthesis, of those measuring a relevant outcome, three out of nine studies measuring interventions targeted at people with two physical health conditions found positive effects (33%), two of which were

**Table 4** Pooled effects of outcomes, stratified by intervention length

| Post-intervention | <3 months | | | 3-6 months | | | >6 months | | |
|---|---|---|---|---|---|---|---|---|---|
| | n | SMD (95%CI) | I² | n | SMD (95%CI) | I² | n | SMD (95%CI) | I² |
| Quality of life | 3 | 0.38 (−0.38 to 1.13) | 81% | 1 | N/A | N/A | 4 | 0.11 (−0.13 to 0.35) | 55% |
| Clinical endpoints | 0 | N/A | N/A | 4 | −0.33 (−1.30 to 0.65) | 96% | 7 | **−0.15 (−0.28 to −0.02)** | 44% |
| Behaviour change | 2 | N/A | N/A | 2 | N/A | N/A | 2 | N/A | N/A |
| Functioning | 1 | N/A | N/A | 8 | 0.07 (−0.02 to 0.16) | 0% | 5 | 0.07 (−0.15 to 0.30) | 65% |
| Psychological distress (depression) | 6 | **−0.36 (−0.57 to −0.14)** | 7% | 10 | −0.42 (−0.96 to 0.12) | 97% | 5 | **−0.50 (−0.69 to −0.30)** | 62% |
| Psychological distress (anxiety) | 3 | −0.20 (−0.65 to 0.26) | 50% | 4 | −0.09 (−0.19 to 0.01) | 0% | 2 | N/A | N/A |
| Emotional well-being | 2 | N/A | N/A | 3 | 0.34 (−0.28 to 0.96) | 90% | 2 | N/A | N/A |
| Maintenance | | | | | | | | | |
| Functioning | 1 | N/A | N/A | 6 | 0.03 (−0.10 to 0.17) | 31% | 1 | N/A | N/A |
| Psychological distress (depression) | 2 | N/A | N/A | 3 | 0.26 (−0.67 to 1.20) | 96% | 3 | **−0.28 (−0.46 to −0.10)** | 94% |
| Emotional well-being | 2 | N/A | N/A | 5 | **0.28 (0.02 to 0.53)** | 81% | 0 | N/A | N/A |

n is number of studies measuring the outcome. P value is considered significant if ≤0.05.
Positive results for psychological distress and clinical endpoints, indicated by a decrease in scores and for emotional well-being by an increase in scores.
N/A, not available; SMD, standardised mean difference.

measuring clinical endpoints. Of those measuring effects in people with a physical and mental health condition, three out of six studies found positive effects (50%), and these were all measuring psychological distress (depression) (see online supplemental table 3).

## DISCUSSION

Across the 53 included studies, types of complex interventions, LTCs targeted and outcome measures used varied widely. Overall, there is evidence that complex interventions can improve the outcomes of people with M-LTCs, and the outcome measures most sensitive to change were emotional well-being and psychological distress.

In subgroup analyses, interventions that lasted for longer periods of time and had a collaborative care approach were associated with greater beneficial outcomes than shorter interventions with a self-management focus. Interventions with a cognitive and/or behavioural activation approach and interventions targeted at people with a combination of mental and physical health conditions had better outcomes when measuring psychological distress (depression) measures.

### Results in context

Developing interventions for people with single LTCs centred around psychological outcomes is not a new concept, with psychological distress and emotional well-being factors being the most amenable to change.[21] This is particularly important when exploring psychological factors, such as psychological capability, which enable a person with a health condition to believe they can engage in the necessary behaviour to improve their outcomes.[22] In the absence of psychological capability, an intervention targeting this, through goal setting, problem solving and self-efficacy, could improve well-being outcomes.[23] These concepts for people with single LTCs appear to transfer over to those with M-LTCs, with psychological distress measures showing a significant improvement in those with a combination of physical and mental health conditions. This is perhaps unsurprising, given the likely higher psychological burden placed on those living with M-LTCs.

In people with single mental LTCs, cognitive and/or behavioural activation-centred interventions have a strong evidence base from systematic reviews for reducing peoples' psychological distress and depression symptoms.[24 25] The findings from this review suggest these interventions have similar improvements for psychological distress in people with a combination of physcial and mental LTCs. The outcomes for this intervention type showed low heterogeneity, which may be due to greater standardised of psychological therapy approaches, which are often manualised with regular supervision. Despite this, often quality of life is chosen as the primary outcome to test the effectiveness of complex interventions, when single clinical outcomes cannot be used. However, here, we have shown that emotional well-being or psychological

distress may be the outcomes more able to detect a change in people with M-LTCs. We have further explored the intervention components (behaviour change techniques) and how they link to intervention effectiveness, which is currently under review for publication.

Looking at the outcomes, the most effective intervention type for people with M-LTCs appears to be collaborative care. Collaborative care involves integrating health and care services across specialties and services to improve the outcomes of the people they are caring for.[26] Other reviews have found positive effects of collaborative care interventions for people with a combination of at least one physical LTC and depression.[26 27] The mechanisms that underpin this change are thought to be twofold, at the individual level, with the person being able to manage their physical LTC better once the depressive symptoms have been reduced, and at the institutional level, with improved delivery of care with a patient centred focus.[26] However, heterogeneity was still substantial for this outcome, possibly as collaborative care interventions can involve differing configurations of healthcare professionals and contact frequencies which may impact effectiveness. Further work needs to explore the most effective components of collaborative care interventions when greater numbers of trials are available.

Alongside the type of intervention, the length of the intervention also seems to be an important factor for effectiveness, with interventions lasting for longer than 6 months improving outcomes postintervention. In line with guidance, when designing an intervention, the minimum clinically effective dose should be identified and empirically tested.[8 28] There is no evidence suggesting that length and the effectiveness of interventions are linked in people with single LTCs. However, it may be that the complex and interacting needs of people with M-LTCs require longer interventions to ensure that the skills to self-manage and the systems to support the individual are in place.[12] Also, it is worth noting that even with interventions lasting for longer than 6 months, maintenance effects are only seen for psychological distress measures, highlighting again that these outcomes are the most amenable to change[21] and that if we want other outcomes to continue to be improved, then support beyond the intervention period is probably needed. The variable heterogeneity in the intervention length analysis may be due to differing intervention types; however, we lacked sufficient studies to explore the effect of length within the different intervention types.

Alongside these findings, it is worth highlighting the lack of studies exploring some of the outcomes of interest, including pain, clinical endpoints and behaviour change. Therefore, the lack of significant effects for these may be due to the lack of studies measuring them as an outcome. This is even more apparent when looking at the maintenance data.

## Future research

The findings presented here highlight the intervention type and length that are associated with improved outcomes in people with M-LTCs. Therefore, future behaviour change interventions should explore collaborative care approaches that last for longer than 6 months. In addition to this, while postintervention outcomes are effective in those lasting more than 6 months, findings around maintenance effects are less conclusive. This suggests more research is needed to determine how to ensure effective interventions have long-lasting effects and improvements can be maintained over time once intervention periods have ended. As collaborative care and self-management intervention types had high heterogeneity, future work should explore the most effective components within these, perhaps utilising methods such as component network meta-analysis.

In addition, while the findings here suggest that interventions to improve psychological distress and emotional well-being outcomes in those with mental LTCs have been successfully applied to those with mental and physical LTCs, the results for those with two or more physical LTCs are less conclusive. Future research needs to explore why this might be, and whether different approaches are needed, with specific intervention components, to address the needs of people with multiple physical LTCs, alongside identifying which outcomes are most appropriate to measure this mechanism of action.[29]

## Limitations

Double screening of eligible papers was completed for a percentage of the identified studies, meaning some eligible papers could have been missed. However, through piloting, good consensus was found and the lead reviewer (TAR) was most familiar with the eligibility criteria, the papers of interest and the process of screening.

Complex interventions are difficult to define and therefore, determining whether a study was eligible based on this criterion and grouping interventions for subgroup meta-analysis can be a challenge. To overcome this potential bias, if there was any doubt, studies were included and discussed with a second (DN) and sometimes a third (RF) reviewer to establish consensus about eligibility and grouping, to make this process as consistent as possible.

Despite using a random-effects model, many of the meta-analyses results had high heterogeneity, suggesting that they may not have been comparable. Preplanned subgroup analysis was conducted to explore this. For psychological distress, the cognitive and/or behavioural activation intervention type and grouping interventions lasting for longer than 6 months and into physical-physical health conditions appeared to lower heterogeneity. Other sources of heterogeneity may be the different outcome measures used for each outcome across the studies. Therefore, the analyses pooling other intervention types and conditions should be viewed with caution, and indicative of areas for further research, rather than as definitive.

Searches were conducted over a year prior to submission. A full updated search was not possible, due to team resources and funding. The search was rerun in Medline and an additional 356 papers were identified and a brief screen found one paper which may have been eligible.[30] This intervention focused on behaviour change in people with depression and type 2 diabetes and found positive outcomes for clinical outcomes and psychological outcomes, which confirm the conclusions of this review and meta-analysis.

## CONCLUSIONS

In conclusion, effective complex interventions to improve one or more outcomes of people with M-LTCs are possible. For those with a combination of physical and mental LTCs, a cognitive and/or behavioural activation approach may be best, and emotional well-being and psychological distress outcomes are most likely to detect change and effectiveness. Evidence around those with multiple physical LTCs is less clear, however, a collaborative care approach is likely the most appropriate, but the intervention components and outcome to measure this still need to be identified to ensure the best outcomes for people with M-LTCs.

**Contributors** TAR (guarantor): principal investigator for the study, developed the protocol, conducted the searches, screened all the abstracts and full texts, completed outcome measure data extraction, conducted meta-analysis and narrative synthesis, wrote and finalised the manuscript. DN: coinvestigator for the study, second reviewer for screening and quality assessment, supported with analysis and interpretation and reviewed the final manuscript. RF: supervised TAR through the process, supported with development of project and protocol, acted as third reviewer when consensus could not be agreed, supported with interpretation of meta-analysis, writing up the results, and reviewed the final manuscript. MA: supported with development of project and protocol and with the interpretation of findings and reviewed the final manuscript. LD: completed data extraction for all studies included and reviewed the final manuscript. JR: supported with development of project and protocol and reviewed the final manuscript. JH: public contributor assisting with project design, finalising the research questions and identifying outcome measures of importance. MM: public contributor assisting with project design, finalising the research questions and identifying outcome measures of importance. ST: supported with development of project and protocol and reviewed the final manuscript. KW: supervised TAR through the process, supported with development of project, protocol and analysis plan and reviewed the final manuscript.

**Funding** This study/project is funded by the National Institute for Health and Care Research (NIHR) School for Primary Care Research (572).

**Disclaimer** The views expressed are those of the author(s) and not necessarily those of the NIHR or the Department of Health and Social Care.

**Competing interests** None declared.

**Patient and public involvement** Patients and/or the public were involved in the design, or conduct, or reporting, or dissemination plans of this research. Refer to the Methods section for further details.

**Patient consent for publication** Not applicable.

**Provenance and peer review** Not commissioned; externally peer reviewed.

**Data availability statement** Data are available on reasonable request. All data relevant to the study are included in the article or uploaded as online supplemental information.

**ORCID iDs**
Tasmin Alanna Rookes http://orcid.org/0000-0001-6330-7059
Rachael Frost http://orcid.org/0000-0003-3523-0052
Kate Walters http://orcid.org/0000-0003-2173-2430

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
