## [Reviewer comments · BMJ Open]

ARTICLE DETAILS

TITLE (PROVISIONAL)	The effectiveness of complex behaviour change interventions tested in randomised controlled trials for people with multiple long-term conditions (M-LTCs): systematic review with meta-analysis
AUTHORS	Rookes, Tasmin; Nimmons, Danielle; Frost, Rachael; Armstrong, Megan; Davies, Laura; Ross, Jamie; Hopkins, Jane; Mistry, Manoj; Taylor, Stephanie; Walters, Kate

VERSION 1 – REVIEW

REVIEWER	Bierman, Arlene Agency for Healthcare Research and Quality
REVIEW RETURNED	09-Dec-2023

GENERAL COMMENTS	The authors conducted a systematic review of behavior change interventions designed to improve outcomes in people living with multiple chronic conditions. This is an important topic, and the synthesis of the literature could be of help to clinicians and health systems seeking to improve care for this growing population, as well as researchers seeking to build on this work. In its current form the manuscript raises a number of significant concerns and could be strengthened by addressing the comments below. 1. The introduction discusses complex interventions for people living with MCC broadly but does not include discussion of the reviews aim of examining complex behavioral interventions.2. The secondary aim of identifying measures sensitive to change appears to be circular. Lack of effect could signify a need to modify the intervention. Measures of the same construct may vary in responsiveness and in the case the most responsive measure could be selected. However, measures overall should be chosen based on assessing the specific aims of the study.3. "This cutoff date was chosen, to align with the dates of papers identified in a previous Cochrane systematic review for complex interventions in people with M-LTCs". Please provide the rationale for aligning with the Cochrane review, and explicitly state what this review adds to the Cochrane review. The Cochrane reference in the paper is from 2016, although an updated review is available from 2021.4. The abstract states only RCTs were included. This should also be stated in the methods section.5. No cutoff sample size was required for inclusion, and several small pilot studies are included in the review. How would the possibility that they were not powered to detect change on included measures effect results and their interpretation.6. No criteria off for duration of intervention was included, and included studies were had durations as short as three weeks, unlikely to be long enough to effect outcomes.
---

	7. Definition of collaborative care, “Interventions targeting the participants, the clinical team, and the system they are embedded within.” Is broad and should be refined to specify what constituted collaborative care. Collaborative care may include behavior change interventions, but collaborative care in and of itself is not a behavior change intervention. The Cochrane review discusses interprofessional collaboration not collaborative care, and there are differences between the two. 8. RCTs included multicomponent interventions no information is included on intervention components. A table describing the intervention components and how these were used in combination would be of value. 9. It would be helpful if the narrative synthesis included more information about the interventions tested. 10. How might the high degree of heterogeneity of the studies influenced results? 11. In the discussion clearly articulate what this review adds to the literature, and what is consistent with prior studies. 12. A careful read for copyediting and clarity is needed. For example, from the intro “Managing (sp.)M-LTCs presents a different challenge (than managing) to those with single LTCs due to the often-fragmented care across multiple primary and secondary care teams and the varied and competing needs of the individual.”
--	--

REVIEWER	van den Akker, Marjan University of Frankfurt, Institute of General Practice
REVIEW RETURNED	16-Jan-2024

GENERAL COMMENTS	BMJOpen-2023-081104 Dear Editor, Thank you for giving me the opportunity to read this interesting paper on a timely and relevant topic. I have some questions and comments, which hopefully can contribute to the further improvement of this manuscript. 1. I assume Figure 1 was not actually meant to be placed in the abstract, but in the main text instead? 2. In the Search example in Suppl.File 1, it becomes clear, that the authors used an explicit list of chronic diseases, that had to be included in title or abstract: why this restriction? How does this affect the number of hits before screening? 3. Time frame (p.4, line 27): published until January 2022; this is almost 2 years before submission. Did the authors consider a search update resp. I think the authors should have performed a search update. 4. At the bottom of page 6 it is mentioned that none of studies used emotional well-being as the primary outcome. Just a few lines below, the first result of the meta-analysis reported, concerns emotional well-being, probably a secondary outcome? I would suggest to report the main secondary outcomes in the paragraph Outcome measures, or alternatively to distinguish between primary and secondary outcomes in the results. Also, in the discussion this might be further elaborated. 5. I would suggest to reorganize table 2 in such a way, that the columns concerning the maintenance are placed besides (instead of below) those concerning post-intervention. This gives the reader an easier overview of the results. 6. The title of Table 3 could be more self-explaining, when including the information regarding follow-up. 7. I have a few questions regarding the pooling procedures that were used; the methodological description of the meta-analysis /
--

	statistical pooling is very brief and could benefit from more details and motivation of choices:  • why did the authors choose for a random effects model? According to the Cochrane Manual decision about the model should be made before inspection of the studies' results. • It is not considered valuable to investigate heterogeneity when there are only very few studies (again according to the Cochrane manual). • For the few outcomes that have a considerable number of underlying studies (e.g. depression post-intervention), a further analysis of the heterogeneity is needed. • It is important to further explore the variation in results, in particular if there is inconsistency in the direction of results. 8. In my opinion, the discussion could pay more attention to the overall scarcity of studies for some of the results, as well as the lack of results on many of the endpoints, e.g. clinical endpoints, behavior change and functioning. 9. To improve understanding of the results, I would appreciate one or two additional columns in Supplementary Table 2, which briefly presents the results for each of the studies. Did the authors try to look for patterns or similarities in interventions among successful trials, in particular in the most promising category of collaborative care?
--	---

VERSION 1 – AUTHOR RESPONSE

1	The introduction discusses complex interventions for people living with MCC broadly but does not include discussion of the reviews aim of examining complex behavioral interventions.	Text added to the introduction with recently published systematic review: “In a recent systematic review and meta-analysis of 14 studies testing behavioural interventions in people with M-LTCs, had little to no effect on physical activity and weight loss (11).” Page 2, paragraph 4.
1	The secondary aim of identifying measures sensitive to change appears to be circular. Lack of effect could signify a need to modify the intervention. Measures of the same construct may vary in responsiveness and in the case the most responsive measure could be selected. However, measures overall should be chosen based on assessing the specific aims of the study.	I understand this argument, however, we have purposively not just included the primary outcome measures. You can see from the “outcome characteristics” in the results section what the primary outcomes are, and you can see that no studies measure emotional wellbeing as a primary outcome, but we have found this when you pool the secondary outcomes then this is very amenable to change. We have added in a section about secondary outcomes to make this point clearer. Page 6, paragraph 3.
1	“This cutoff date was chosen, to align with the dates of papers identified in a previous Cochrane systematic review for complex interventions in people with M-LTCs”. Please provide the rationale for aligning with the Cochrane review, and explicitly state what this review adds to the Cochrane review. The Cochrane reference in the	The wording of this has been updated to be clearer. It is not an update of this review, but the Smith review used a similar search strategy for interventions and the oldest included intervention was from 1999, which is why we used this as the oldest date. Page 3, paragraph 4.

	paper is from 2016, although an updated review is available from 2021.	
1	The abstract states only RCTs were included. This should also be stated in the methods section.	Added to the inclusion criteria in the methods section.
1	No cutoff sample size was required for inclusion, and several small pilot studies are included in the review. How would the possibility that they were not powered to detect change on included measures effect results and their interpretation.	The sample size was included in the meta-analysis, so they were weighted in the analysis.
1	No criteria off for duration of intervention was included, and included studies were had durations as short as three weeks, unlikely to be long enough to effect outcomes.	We agree with this, which is what led us to do sub-group analysis based on duration of intervention in section 2. This led to one of our main conclusions that interventions that lasted for longer than 6-months to be most effective.
1	Definition of collaborative care, “Interventions targeting the participants, the clinical team, and the system they are embedded within.” Is broad and should be refined to specify what constituted collaborative care. Collaborative care may include behavior change interventions, but collaborative care in and of itself is not a behavior change intervention. The Cochrane review discusses interprofessional collaboration not collaborative care, and there are differences between the two.	All the interventions included a behaviour change component, as this was our inclusion criteria. Within these interventions we then identified 3 sub-categories with additional components, which was what made them complex. I have added in that they were behaviour change interventions that targeted the 3 areas for clarity in table 1.
1	RCTs included multicomponent interventions no information is included on intervention components. A table describing the intervention components and how these were used in combination would be of value.	We have generated another paper focussing on the behaviour change techniques, which has also been submitted for publication. This information is not presented here, as it would make the paper too long and would lead to duplication. We have added a statement linking to this in the discussion. Page 11, paragraph 4.
1	It would be helpful if the narrative synthesis included more information about the interventions tested.	This information is in supplementary table 3.
1	How might the high degree of heterogeneity of the studies influenced results?	This is discussed in the limitations section of the discussion.
1	In the discussion clearly articulate what this review adds to the literature, and what is consistent with prior studies.	Some more detail around impact has been added into the future research section. Page 12, paragraph 2.
1	A careful read for copyediting and clarity is needed. For example, from the intro “Manging (sp.)M-LTCs presents a different challenge (than	This has been done and edits have been made for these errors.

	managing) to those with single LTCs due to the often-fragmented care across multiple primary and secondary care teams and the varied and competing needs of the individual.”	
2	I assume Figure 1 was not actually meant to be placed in the abstract, but in the main text instead?	This has been removed and the results added to the text of the abstract. Forest plots have been added to supplementary figure 3.
2	In the Search example in Suppl.File 1, it becomes clear, that the authors used an explicit list of chronic diseases, that had to be included in title or abstract: why this restriction? How does this affect the number of hits before screening?	We included terms around multimorbidity and individual conditions to ensure both forms of wording were identified in the search. Authors will almost always include these features about the population in the title and abstract. Given the large number of hits returned and included we think this has been sufficient to capture all the relevant articles.
2	Time frame (p.4, line 27): published until January 2022; this is almost 2 years before submission. Did the authors consider a search update resp. I think the authors should have performed a search update.	Unfortunately, due to the funding period finishing and current resources it is not possible to do a full update of the search. A recent published piece highlights that for complex multi-component reviews which take a long time to complete there is likely to be less comprise from having a slightly outdated search (https://ebm.bmj.com/content/ebmed/early/2022/12/08/bmjebm-2022-12060.full.pdf). We have rerun the search in Medline, which identified an additional 356 number of results and through a quick screen identified only 1 study that would have been eligible and the results support our findings and we do not expect any change will happen adding this study. Some detail has been added to the discussion – limitations section around this.
2	At the bottom of page 6 it is mentioned that none of studies used emotional well-being as the primary outcome. Just a few lines below, the first result of the meta-analysis reported, concerns emotional well-being, probably a secondary outcome? I would suggest to report the main secondary outcomes in the paragraph Outcome measures, or alternatively to distinguish between primary and secondary outcomes in the results. Also, in the discussion this might be further elaborated.	A section has been added to the outcome characteristics section to make it clear that these were also included in the meta-analysis and which ones these were. Page 6, paragraph 3.
2	I would suggest to reorganize table 2 in such a way, that the columns concerning the maintenance are	This has been edited and added in the suggested format.

	placed besides (instead of below) those concerning post-intervention. This gives the reader an easier overview of the results.	
2	The title of Table 3 could be more self-explaining, when including the information regarding follow-up.	This has been updated to be clearer.
2	I have a few questions regarding the pooling procedures that were used; the methodological description of the meta-analysis / statistical pooling is very brief and could benefit from more details and motivation of choices:  • why did the authors choose for a random effects model? According to the Cochrane Manual decision about the model should be made before inspection of the studies' results. • It is not considered valuable to investigate heterogeneity when there are only very few studies (again according to the Cochrane manual). • For the few outcomes that have a considerable number of underlying studies (e.g. depression post-intervention), a further analysis of the heterogeneity is needed. • It is important to further explore the variation in results, in particular if there is inconsistency in the direction of results. 	 1. Random effects model: due to the heterogeneity of the samples and the interventions we thought this would be the most appropriate model to use. This decision was made prior to conducting any analyses. Page 4, paragraph 3. 2. Heterogeneity: This was only explored and discussed for studies with effect results/with large samples. The I2 scores have just been included in the tables for transparency but can be removed if preferred. 3. The heterogeneity has been discussed throughout in terms of the types of interventions and the populations that have been targeted in each study and this has been added as a limitation. 4. There is only one result which is inconsistent behaviour change for interventions less than 6-months, but this is likely due to chance, as only 2 studies have been included, so little emphasis has been placed on this.
2	In my opinion, the discussion could pay more attention to the overall scarcity of studies for some of the results, as well as the lack of results on many of the endpoints, e.g. clinical endpoints, behaviour change and functioning.	Thank you for raising this point, some detail around this has been added to the discussion. Page 11, paragraph 1.
2	To improve understanding of the results, I would appreciate one or two additional columns in Supplementary Table 2, which briefly presents the results for each of the studies. Did the authors try to look for patterns or similarities in interventions among successful trials, in particular in the most promising category of collaborative care?	We have generated another paper, which heavily references this paper, focussing on the behaviour change techniques, which has also been submitted for publication. This information is not presented here, as it would make the paper too long and would lead to duplication. We have added a statement linking to this in the discussion. Page 11, paragraph 4.

VERSION 2 – REVIEW

REVIEWER	van den Akker, Marjan University of Frankfurt, Institute of General Practice
REVIEW RETURNED	21-Feb-2024

GENERAL COMMENTS	I'm satisfied with the changes the authors have made and have not further comments.
---

REVIEWER	Bierman, Arlene Agency for Healthcare Research and Quality
REVIEW RETURNED	09-Dec-2023

GENERAL COMMENTS	The authors conducted a systematic review of behavior change interventions designed to improve outcomes in people living with multiple chronic conditions. This is an important topic, and the synthesis of the literature could be of help to clinicians and health systems seeking to improve care for this growing population, as well as researchers seeking to build on this work. In its current form the manuscript raises a number of significant concerns and could be strengthened by addressing the comments below.  1. The introduction discusses complex interventions for people living with MCC broadly but does not include discussion of the reviews aim of examining complex behavioral interventions. 2. The secondary aim of identifying measures sensitive to change appears to be circular. Lack of effect could signify a need to modify the intervention. Measures of the same construct may vary in responsiveness and in the case the most responsive measure could be selected. However, measures overall should be chosen based on assessing the specific aims of the study. 3. "This cutoff date was chosen, to align with the dates of papers identified in a previous Cochrane systematic review for complex interventions in people with M-LTCs". Please provide the rationale for aligning with the Cochrane review, and explicitly state what this review adds to the Cochrane review. The Cochrane reference in the paper is from 2016, although an updated review is available from 2021. 4. The abstract states only RCTs were included. This should also be stated in the methods section. 5. No cutoff sample size was required for inclusion, and several small pilot studies are included in the review. How would the possibility that they were not powered to detect change on included measures effect results and their interpretation. 6. No criteria off for duration of intervention was included, and included studies were had durations as short as three weeks, unlikely to be long enough to effect outcomes. 7. Definition of collaborative care, "Interventions targeting the participants, the clinical team, and the system they are embedded within." Is broad and should be refined to specify what constituted collaborative care. Collaborative care may include behavior change interventions, but collaborative care in and of itself is not a behavior change intervention. The Cochrane review discusses interprofessional collaboration not collaborative care, and there are differences between the two. 8. RCTs included multicomponent interventions no information is included on intervention components. A table describing the intervention components and how these were used in combination would be of value.
--

	9. It would be helpful if the narrative synthesis included more information about the interventions tested. 10. How might the high degree of heterogeneity of the studies influenced results? 11. In the discussion clearly articulate what this review adds to the literature, and what is consistent with prior studies. 12. A careful read for copyediting and clarity is needed. For example, from the intro “Managing (sp.)M-LTCs presents a different challenge (than managing) to those with single LTCs due to the often-fragmented care across multiple primary and secondary care teams and the varied and competing needs of the individual.”
--	--

VERSION 2 – AUTHOR RESPONSE

1	The introduction discusses complex interventions for people living with MCC broadly but does not include discussion of the reviews aim of examining complex behavioral interventions.	Text added to the introduction with recently published systematic review: “In a recent systematic review and meta-analysis of 14 studies testing behavioural interventions in people with M-LTCs, had little to no effect on physical activity and weight loss (11).” Page 2, paragraph 4.
1	The secondary aim of identifying measures sensitive to change appears to be circular. Lack of effect could signify a need to modify the intervention. Measures of the same construct may vary in responsiveness and in the case the most responsive measure could be selected. However, measures overall should be chosen based on assessing the specific aims of the study.	I understand this argument, however, we have purposively not just included the primary outcome measures. You can see from the “outcome characteristics” in the results section what the primary outcomes are, and you can see that no studies measure emotional wellbeing as a primary outcome, but we have found this when you pool the secondary outcomes then this is very amenable to change. We have added in a section about secondary outcomes to make this point clearer. Page 6, paragraph 3.
1	“This cutoff date was chosen, to align with the dates of papers identified in a previous Cochrane systematic review for complex interventions in people with M-LTCs”. Please provide the rationale for aligning with the Cochrane review, and explicitly state what this review adds to the Cochrane review. The Cochrane reference in the paper is from 2016, although an updated review is available from 2021.	The wording of this has been updated to be clearer. It is not an update of this review, but the Smith review used a similar search strategy for interventions and the oldest included intervention was from 1999, which is why we used this as the oldest date. Page 3, paragraph 4.
1	The abstract states only RCTs were included. This should also be stated in the methods section.	Added to the inclusion criteria in the methods section.
1	No cutoff sample size was required for inclusion, and several small pilot studies are included in the review. How would the possibility that they were not powered to detect change on	The sample size was included in the meta-analysis, so they were weighted in the analysis.

	included measures effect results and their interpretation.	
1	No criteria off for duration of intervention was included, and included studies were had durations as short as three weeks, unlikely to be long enough to effect outcomes.	We agree with this, which is what led us to do sub-group analysis based on duration of intervention in section 2. This led to one of our main conclusions that interventions that lasted for longer than 6-months to be most effective.
1	Definition of collaborative care, “Interventions targeting the participants, the clinical team, and the system they are embedded within.” Is broad and should be refined to specify what constituted collaborative care. Collaborative care may include behavior change interventions, but collaborative care in and of itself is not a behavior change intervention. The Cochrane review discusses interprofessional collaboration not collaborative care, and there are differences between the two.	All the interventions included a behaviour change component, as this was our inclusion criteria. Within these interventions we then identified 3 sub-categories with additional components, which was what made them complex. I have added in that they were behaviour change interventions that targeted the 3 areas for clarity in table 1.
1	RCTs included multicomponent interventions no information is included on intervention components. A table describing the intervention components and how these were used in combination would be of value.	We have generated another paper focussing on the behaviour change techniques, which has also been submitted for publication. This information is not presented here, as it would make the paper too long and would lead to duplication. We have added a statement linking to this in the discussion. Page 11, paragraph 4.
1	It would be helpful if the narrative synthesis included more information about the interventions tested.	This information is in supplementary table 3.
1	How might the high degree of heterogeneity of the studies influenced results?	This is discussed in the limitations section of the discussion.
1	In the discussion clearly articulate what this review adds to the literature, and what is consistent with prior studies.	Some more detail around impact has been added into the future research section. Page 12, paragraph 2.
1	A careful read for copyediting and clarity is needed. For example, from the intro “Manging (sp.)M-LTCs presents a different challenge (than managing) to those with single LTCs due to the often-fragmented care across multiple primary and secondary care teams and the varied	This has been done and edits have been made for these errors.

	and competing needs of the individual.”	
2	I assume Figure 1 was not actually meant to be placed in the abstract, but in the main text instead?	This has been removed and the results added to the text of the abstract. Forest plots have been added to supplementary figure 3.
2	In the Search example in Suppl.File 1, it becomes clear, that the authors used an explicit list of chronic diseases, that had to be included in title or abstract: why this restriction? How does this affect the number of hits before screening?	We included terms around multimorbidity and individual conditions to ensure both forms of wording were identified in the search. Authors will almost always include these features about the population in the title and abstract. Given the large number of hits returned and included we think this has been sufficient to capture all the relevant articles.
2	Time frame (p.4, line 27): published until January 2022; this is almost 2 years before submission. Did the authors consider a search update resp. I think the authors should have performed a search update.	Unfortunately, due to the funding period finishing and current resources it is not possible to do a full update of the search. A recent published piece highlights that for complex multi-component reviews which take a long time to complete there is likely to be less comprise from having a slightly outdated search (https://ebm.bmj.com/content/ebmed/early/2022/12/08/bmjebm-2022-12060.full.pdf). We have rerun the search in Medline, which identified an additional 356 number of results and through a quick screen identified only 1 study that would have been eligible and the results support our findings and we do not expect any change will happen adding this study. Some detail has been added to the discussion – limitations section around this.
2	At the bottom of page 6 it is mentioned that none of studies used emotional well-being as the primary outcome. Just a few lines below, the first result of the meta-analysis reported, concerns emotional well-being, probably a secondary outcome? I would suggest to report the main secondary outcomes in the paragraph Outcome measures, or alternatively to distinguish between primary and secondary outcomes in the results. Also, in the discussion this might be further elaborated.	A section has been added to the outcome characteristics section to make it clear that these were also included in the meta-analysis and which ones these were. Page 6, paragraph 3.
2	I would suggest to reorganize table 2 in such a way, that the columns	This has been edited and added in the suggested format.

	concerning the maintenance are placed besides (instead of below) those concerning post-intervention. This gives the reader an easier overview of the results.	
2	The title of Table 3 could be more self-explaining, when including the information regarding follow-up.	This has been updated to be clearer.
2	I have a few questions regarding the pooling procedures that were used; the methodological description of the meta-analysis / statistical pooling is very brief and could benefit from more details and motivation of choices:  • why did the authors choose for a random effects model? According to the Cochrane Manual decision about the model should be made before inspection of the studies' results. • It is not considered valuable to investigate heterogeneity when there are only very few studies (again according to the Cochrane manual). • For the few outcomes that have a considerable number of underlying studies (e.g. depression post-intervention), a further analysis of the heterogeneity is needed. • It is important to further explore the variation in results, in particular if there is inconsistency in the direction of results. 	 1. Random effects model: due to the heterogeneity of the samples and the interventions we thought this would be the most appropriate model to use. This decision was made prior to conducting any analyses. Page 4, paragraph 3. 2. Heterogeneity: This was only explored and discussed for studies with effect results/with large samples. The I2 scores have just been included in the tables for transparency but can be removed if preferred. 3. The heterogeneity has been discussed throughout in terms of the types of interventions and the populations that have been targeted in each study and this has been added as a limitation. 4. There is only one result which is inconsistent behaviour change for interventions less than 6-months, but this is likely due to chance, as only 2 studies have been included, so little emphasis has been placed on this.
2	In my opinion, the discussion could pay more attention to the overall scarcity of studies for some of the results, as well as the lack of results on many of the endpoints, e.g. clinical endpoints, behaviour change and functioning.	Thank you for raising this point, some detail around this has been added to the discussion. Page 11, paragraph 1.
2	To improve understanding of the results, I would appreciate one or two additional columns in Supplementary Table 2, which briefly presents the results for each of the studies. Did the authors try to look for patterns or similarities in interventions among	We have generated another paper, which heavily references this paper, focussing on the behaviour change techniques, which has also been submitted for publication. This information is not presented here, as it would make the paper too long and would lead to duplication. We have

	successful trials, in particular in the most promising category of collaborative care?	added a statement linking to this in the discussion. Page 11, paragraph 4.
--	--	--

VERSION 3 – REVIEW

REVIEWER	van den Akker, Marjan University of Frankfurt, Institute of General Practice
REVIEW RETURNED	21-Feb-2024

GENERAL COMMENTS	I'm satisfied with the changes the authors have made and have not further comments.
---

REVIEWER	Mathes, Tim Universitätsmedizin Göttingen
REVIEW RETURNED	22-Mar-2024

GENERAL COMMENTS	The statistical methods are not described in sufficient detail. Many important aspects are missing. This is:  - The type of random effect model - Handling missing data - Information on how the decision was made on combining probably heterogeneous interventions - Information on how the decision was made to summarize different scales within one SMD The major problem with the analyses/results is that for almost all outcomes high heterogeneity is present but the authors make no sufficient attempts to explore and clarify this heterogeneity or the results should not be pooled. Component-Network-Meta-Analysis might be an approach to resolve heterogeneity. Considering the heterogeneity and the risk of bias of the included studies I think the conclusion is probably too optimistic. I would recommend using GRADE to arrive at a well-rounded conclusion.
--

VERSION 3 – AUTHOR RESPONSE

Reviewer 3	The statistical methods are not described in sufficient detail. Many important aspects are missing. This is:  - The type of random effect model - Handling missing data - Information on how the decision was made on combining probably heterogeneous interventions - Information on how the decision was made to summarize different scales within one SMD 	Additional detail has been added to the text:  - Random effects model (page 4, paragraph 3) - Missing data (Page 3, paragraph 4) - Combining probable heterogeneous results (page 4, paragraph 3) - SMD (page 4, paragraph 2)
Reviewer 3	The major problem with the analyses/results is that for almost all outcomes high	We have removed the results and interpretation of results for

	heterogeneity is present but the authors make no sufficient attempts to explore and clarify this heterogeneity or the results should not be pooled. Component-Network-Meta-Analysis might be an approach to resolve heterogeneity. Considering the heterogeneity and the risk of bias of the included studies I think the conclusion is probably too optimistic. I would recommend using GRADE to arrive at a well-rounded conclusion.	outcomes with only 1 or 2 studies included to make interpretation easier and to be in line with what was stated in the methods. We have more clearly explained the likely impact of heterogeneity on interpretation in the results and discussion. The language around the conclusions has been softened to reflect the potential bias and heterogeneity. As the review aimed to explore the measures most likely to detect change and provides avenues for future research, a GRADE approach was not deemed appropriate as this aims to provide specific recommendations to inform guidelines. Component network meta-analysis was beyond the scope and aims of this review, however, remains a valuable area for future reviews. We have identified this in the section on further research.
--	--	--